# Conservation of Genetic Diversity of Scots Pine (*Pinus sylvestris* L.) in a Central European National Park Based on cpDNA Studies

Paweł Przybylski [1,*], Anna Tereba [1], Joanna Meger [2], Iwona Szyp-Borowska [1] and Łukasz Tyburski [3]

1   Forest Research Institute, Braci Leśnej 3, Sękocin Stary, 05-090 Raszyn, Poland; A.Tereba@ibles.waw.pl (A.T.); i.borowska@ibles.waw.pl (I.S.-B.)
2   Department of Genetics, Kazimierz Wielki University, ul. Chodkiewicza 30, 85-064 Bydgoszcz, Poland; warmbier@ukw.edu.pl
3   Kampinoski National Park, Tetmajera 38, 05-080 Izabelin, Poland; ltyburski@kampinoski-pn.gov.pl
*   Correspondence: p.przybylski@ibles.waw.pl

**Abstract:** In the old pine stands of national parks, it is possible to observe genetic processes in a state free from disturbance by humans. Studies of this type make it possible to evaluate the effectiveness of the conservation of genetic variation and its transfer between generations. The present study was conducted in the largest national forest park in Poland, located in the Central European pine area. The oldest stands of Kampinos National Park and their natural descendants were selected for detailed analyses. The main objective of the study was to compare the mother pine stand, excluded from forest management, with its progeny generations on the basis of their chloroplast DNA (cpDNA), which was used as a diagnostic tool. The results demonstrate significant genetic difference between the maternal and progeny generations of the studied sites. The degree of variation observed in the maternal generation haplotypes in the present study was found to be reduced in the next generation. A significant proportion of the genetic diversity of the studied stands was also lost in the subsequent progeny generation. The obtained results allow conclusions to be drawn about the genetic processes taking place in valuable old-growth forests.

**Keywords:** old growth forests; gene flow; molecular markers; legally protected forests



## 1. Introduction

Scots pine (*Pinus sylvestris* L.), the species investigated in this study, is the second most widespread conifer in the world and has great economic and ecological importance [1]. Pine forests cover 37% of the total global land area and 70% of the land area of the Northern Hemisphere, making pine one of the most important forest-forming trees in the world [1]. Pine is characterised by high morphological and genetic diversity. The maintenance of considerable pine variability is favoured by the transmission of pine genetic information by seeds [2] and pollen [3]. Natural processes affecting the genetic diversity of pine populations can be roughly observed in national parks. In economic stands, forest management leads to the suppression of natural gene flow due to the fragmentation of forest sites, which results from regulation based on forest stand age [4]. The present study was conducted in the largest national forest park in Poland, Kampinos National Park (KNP). Unfortunately, the history of the Kampinos forests indicates periods of the complete deforestation of the national park area influenced by armed conflicts of a global character [5]. On the other hand, the studied ecosystems have been under full legal protection for over 50 years. Therefore, it can be assumed that while the genetic variability of the ancient forests of KNP has been influenced by humans, their regeneration processes are spontaneous. The presented results can be interpreted in the context of adaptation of pine trees to different growth conditions, which follows the results of earlier studies by Przybylski et al. [6].

The results of the presented studies, based on Krzanowska et al. [7] and others, suggest that stands with a high degree of genetic differentiation have a much higher chance of survival and can consequently pass on favourable gene combinations to their offspring. The dynamic changes in climate observed around the world are producing ecosystems that are in a state of dynamic equilibrium in responding to this change. Changes in stands are observed to occur both within and between species [8]. It is undisputed that genetic diversity is important for the long-term survival of species and plays a crucial role in their conservation [9–11]. The genetic diversity of organisms has a major influence on both the adaptive potential of individuals and the response of entire populations to external selection factors [12,13]. However, regulations and policies have long been primarily focused on the more visible elements of biodiversity, such as species or populations, while genetics has been mostly neglected [14]. Ongoing genetic research mainly focuses on stands before and after certain events that have a significant impact on genetic pools, with little work in natural populations under legal protection. The associated analyses usually employ autosomal markers such as microsatellite loci [15,16]. The work presented here replicates the trend of microsatellite DNA sequence analyses, and it should be noted that the study focused on microsatellites in chloroplast DNA (cpDNA). A trend in the study of conifers is the increasing importance of chloroplast DNA analyses (cpSSRs), as observed in, e.g., Fady et al. [17] and Gómez et al. [18]. In Scots pine in Poland, cpDNA has mainly been used to detect hybridisation between *P. sylvestris* × *P. mugo* [19]. Another important scientific study is the description of the complete chloroplast genome of *Pinus uliginosa* (Neumann), which is helpful in explaining the complex taxonomic position of the species [20]. Only a few studies, such as Semerikov et al. [21], have used cpDNA to describe variation in pine stands. The study mentioned above shows a variation between natural populations in Asia and Eastern Europe of 2.1% [21], and slightly lower results were found for populations from Estonia [22]. In Poland, a study of cpDNA diversity between age classes of stocks was conducted [23], which showed a similar level of diversity as between different populations. Wojnicka-Póltorak et al. [23] point to the transfer of foreign pollen to the parent population as one of the reasons for the detected differentiation.

The research presented in this paper focuses on stands excluded from forest management and protected by law in Kampinos National Park, the largest national forest park in Poland. Due to historical events (for example, the First and Second World Wars), it is likely that evolution did not occur by natural regeneration for the stands in the studied areas, in contrast to their progeny. Therefore, the analysis of changes in the gene pool is interesting in terms of the possibility of the intergenerational preservation of genetic diversity and the intensity of gene flow. The formulated hypotheses of the conducted research state that allelic diversity may be preserved in the next generation of a stand. At the same time, it is suspected that pollen from local commercial forest stands influences the formation of the gene pool of the subsequent generation. The main objective of the research presented in this publication is to compare the maternal and progeny generations of pine stands excluded from forest management from the perspective of cpDNA genetic diversity. The obtained results will expand upon basic knowledge regarding the behaviour of pine in natural conditions.

## 2. Materials and Methods

### 2.1. Research Area

The study was conducted in Kampinos National Park (52°19′13″ N, 20°47′23″ E), a location dominated by Scots stands in the upper layer and located in strictly protected areas. The characteristics of the study locations are presented in Table 1 and Figure 1.

**Table 1.** The characteristics of the study locations.

| Location | Czerwińskie Góry | Wilków | Granica | Sieraków | Wiersze | Nart | Krzywa Góra |
|---|---|---|---|---|---|---|---|
| Abbreviation | CG | W | Gr | S | Wi | N | KG |
| Coordinates | 20°23′36.67″ E 52°20′27.693″ N | 20°32′34.005″ E 52°21′45.607″ N | 20°27′50.019″ E 52°17′21.605″ N | 20°46′34.957″ E 52°20′12.144″ N | 20°39′45.706″ E 52°18′34.157″ N | 20°30′2.718″ E 52°17′49.844″ N | 20°25′14.603″ E 52°20′27.744″ N |
| Age * of the dominant *P. sylvestris* | 200–210 (avg.: 205) | 180–200 (avg.: 190) | 160–170 (avg.: 165) | 190–200 (avg.: 195) | app. 160 | 210–230 (avg.: 220) | app. 108 |
| Plant community | *Querco roboris-Pinetum* | *Querco roboris-Pinetum* | SNFPC/*Querco Carpinetum* | *Querco roboris-Pinetum* | *Querco roboris-Pinetum* | *Tilio-Carpinetum* | *Querco roboris-Pinetum* |

\* Unpublished data from Kampinoski National Park.

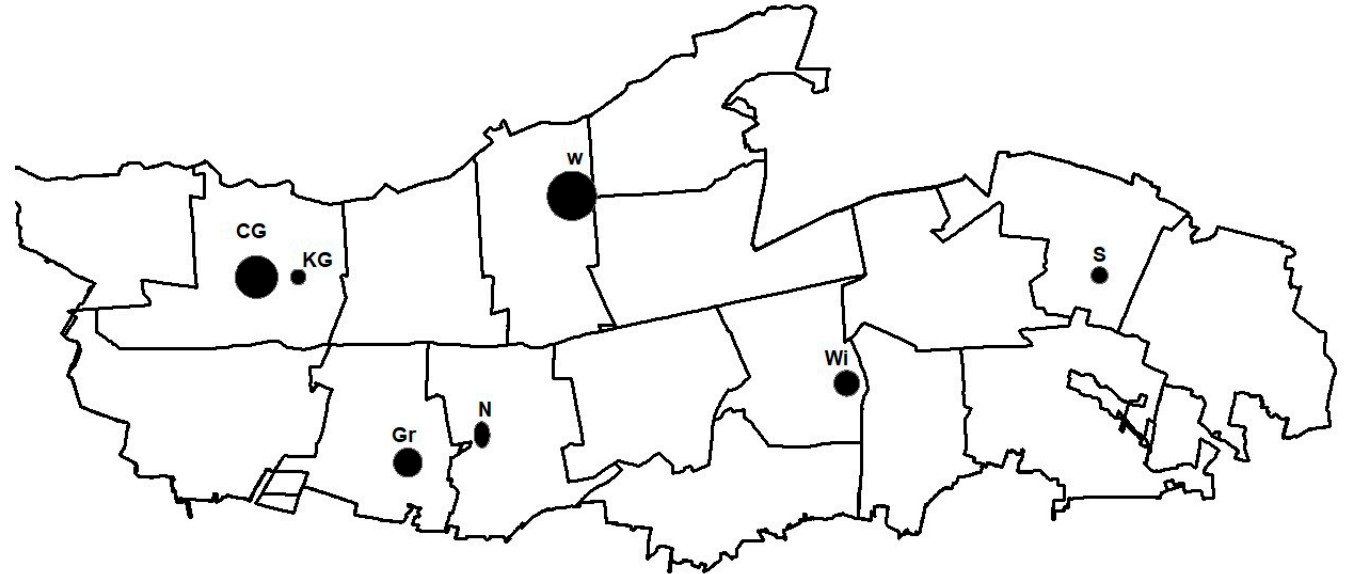

**Figure 1.** Location of the studied stands in the Kampinos NP area. Points mark survey locations based on their area according to the outermost sampled trees. The size of the marked point is based on the differences in the plots. Description according to the abbreviations used.

*2.2. Sample Collection*

Plant material (needles) was collected from each location (Table 1) from 50 randomly selected pines constituting the maternal generation (F1) and 50 randomly selected 1-year-old seedlings, referred to as the progeny generation (F2). The condition for selecting F1 was the distance between samples of at least one height of the dominant stand, while F2 was collected in close proximity to F1 (see Tables S1–S7 in the Supplementary Materials). The planned number of individuals of the F2 generation was not collected for most populations because there were no pine seedlings at the study sites (Tables S1–S7, Supplementary Materials). In addition, the vigour of some samples of the F2 generation was poor. The material in the field was placed in 1.5 mL Eppendorf tubes then was stored in the laboratory under cryogenic conditions (−85 °C).

*2.3. DNA Extraction and Microsatellites Genotyping*

Total genomic DNA was isolated from the collected material using a commercial kit (Macherey–Nagel Gmbh&Co Valencienner Str. 11; 52,355, Dueren, Germany). The quality of the DNA isolate was controlled using 2% agarose gel and a Quawell (LabX, 334 King street, Midland, ON Canada) spectrophotometer. All samples were diluted to 20–30 ng/uL using deionised water. Molecular analyses were performed using 6 (PCP26106; PCP30277; PCP36567; PCP450712; PCP719872; PCP873142) chloroplast microsatellite markers selected on the basis of other research work [24–26]; the forward primers were fluorescently labelled with the fluorochromes VIC, PET, NED, and 6-FAM. Amplification was performed through two multiplex reactions. Each PCR reaction was performed in a volume of 10 μL, with

the following composition: 5 μL Multiplex buffer (Qiagen, Poland), 0.2 μL (10 μM) of each primer, 1 μL of extracted DNA, and PCR-grade water up to a final volume of 10 μL. The PCR thermal profile was as follows: 95 °C for 15 min; followed by 30 cycles at 95 °C for 30 s, 55 °C for 30 s, and 72 ° C for 1 min, with a final extension of 60 °C for 30 min. Genotyping analysis was performed on an ABI 3500 Genetic Analyzer capillary sequencer (Applied Biosystems, Foster City, CA, USA) and allele length analysis was performed using GeneMapper® version 5 (Thermo Fisher Scientific Inc., Carlsbad, CA, USA).

*2.4. Data Analysis*

Haplotypes were determined as a combination of different microsatellite variants across the cpDNA loci. The chloroplast haplotype variation within populations, i.e., the number of haplotypes ($A$), genetic diversity ($He$), haplotype richness ($Rh$), the number of private haplotypes ($P$), and mean genetic distance between individuals ($D^2sh$) were calculated using HAPLOTYPE version 1.05 [27].

Genetic structuring of the cpDNA between and within populations was assessed using analysis of molecular variance (AMOVA), implemented in ARLEQUIN version 3.0 [28], with significance tests based on 10,000 permutations. The distance matrix generated by pairwise $F_{ST}$ values between populations was calculated using ARLEQUIN version 3.0 [25]. The statistical significance of $F_{ST}$ values was assessed using 10,000 permutations. The grouping of analysed stands on the basis of the genetic distance of $N_{ei}$ using Principal Coordinates Analisis (PCoA) as implemented in the GenALEx 6.5 [29] program.

The isolation by distance was tested using the Mantel test [30] of association between the $F_{ST}$-based pairwise genetic distance matrix (i.e., $F_{ij}/(1 - F_{ij})$, where $F_{ij}$ is the $F_{ST}$ for the i-th and the j-th populations) and the matrix of the natural logarithm of geographic distance using GENEPOP version 4.4 software [31]. The significance was assessed with 10,000 random permutations.

## 3. Results

An average of 4.07 alleles were found in F1 generation populations (Table S8 in the Supplementary Materials), with most in the S location (4.5) and the least in CG (3.7). The average effective number of alleles (Ne) was 2.2, between 2.3 and 2.0. Private allele frequencies were described for S (0.5), Wi (0.33), and KG (0.17), while no private alleles were detected for the other populations. The genetic diversity in the 6 stands of the F1 generation was equal to h = 0.5, except for N, where it was 0.4.

For the F2 generation, the average number of alleles was 3.83 (Table S8 in the Supplementary Materials), with the highest in the KG locality (4.2), while the lowest was in S (3.5). The average effective number of alleles (Ne) in the F2 generation was 2.1, between 2.4 and 1.8. Private allele frequencies were indicated for the locations S (0.16), KG (0.33), and Gr (0.33). The genetic diversity in the 6 stands of the F2 generation was equal to h = 0.5, except for S where it was 0.4. The genetic variability of haplotypes between F1 and F2 of the analysed locations is shown in Figure 2. Note that most of the generated haplotypes are identical for the analysed populations. Genetic diversity arises from differences in allele frequencies between populations and generations and from the occurrence of private alleles.

The genetic diversity parameters of the studied locations are summarised in Table 2. At the studied sites, results were obtained for an average of 47.28 plants of the maternal generation (F1) and an average of 34.42 plants of the progeny generation (F2) (Table 2). The determined number of haplotypes ($A$) ranged from 32–44 for F1 and from 15–34 for F2. The lower number of determined F2 haplotypes is related to the number of samples analysed (N) (Table 2). Private haplotypes ($P_h$) were indicated for each site, of which there were much more for F1. The ($P_h$) value was dependent on the haplotype richness ($R_h$). The results showed no differences in genetic diversity ($H_e$) among generations and locations (Table 2). The values of mean genetic distance between individuals ($D^2sh$) were higher for

F1 and particularly high for Sieraków (4.29) and Wiersze (5.52), which were not maintained for F2 (Table 2).

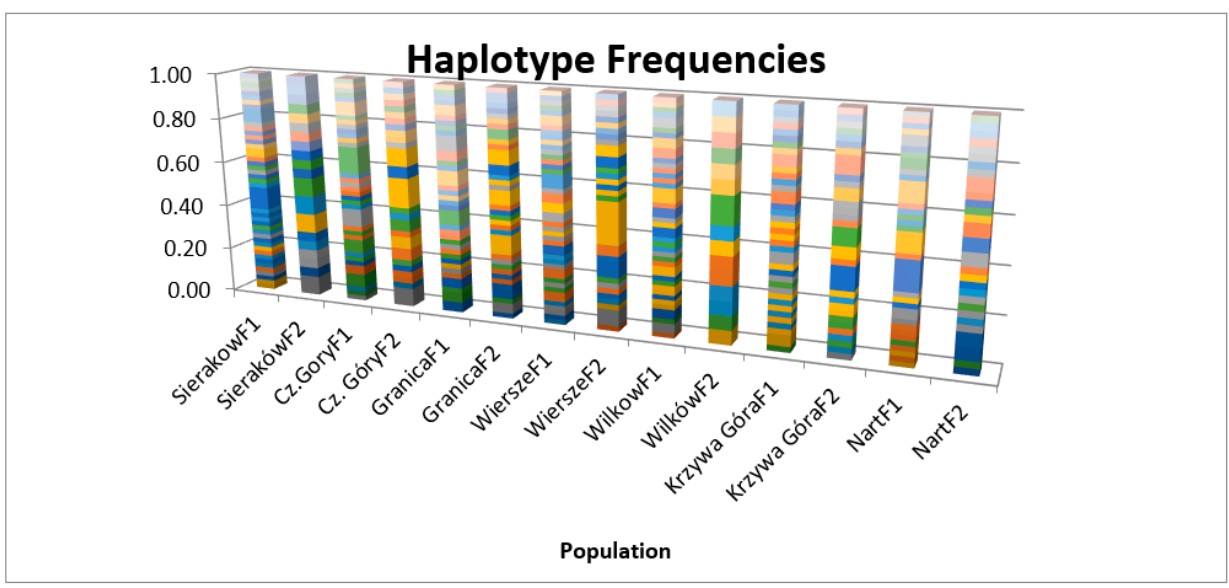

**Figure 2.** Haplotype frequencies expressed as a percentage (*Y*-axis) for the analysed locations of maternal (F1) and progeny (F2) generations. Each colour on the graph corresponds to one of the 254 identified haplotypes.

**Table 2.** Genetic diversity within a population of *P. sylvestris* L. based on 6 cpDNA markers. Number of haplotypes (*A*), the number of private haplotypes (*Ph*), haplotype richness (*Rh*), genetic diversity (*He*), and mean genetic distance between individuals (*D²sh*).

| Population | N | A | $P_h$ | $R_h$ | $H_e$ | $D^2sh$ |
|---|---|---|---|---|---|---|
| F1 generation | | | | | | |
| Sieraków | 50 | 44 | 28 | 35.586 | 0.992 | 4.297 |
| Cz. Góry | 50 | 36 | 19 | 29.532 | 0.976 | 3.625 |
| Granica | 46 | 35 | 19 | 31.015 | 0.986 | 3.218 |
| Wiersze | 49 | 41 | 23 | 34.303 | 0.992 | 5.523 |
| Wilków | 49 | 40 | 18 | 33.606 | 0.991 | 2.754 |
| Krzywa Góra | 41 | 37 | 17 | 36.000 | 0.995 | 3.594 |
| Nart | 46 | 32 | 13 | 28.145 | 0.971 | 2.584 |
| Mean | 47.286 | 37.857 | 19.571 | 32.598 | 0.986 | 3.657 |
| F2 generation | | | | | | |
| Sieraków | 24 | 20 | 11 | 13.261 | 0.986 | 2.557 |
| Cz. Góry | 38 | 27 | 14 | 12.642 | 0.977 | 2.893 |
| Granica | 46 | 34 | 13 | 13.292 | 0.985 | 2.783 |
| Wiersze | 43 | 30 | 13 | 12.123 | 0.961 | 2.508 |
| Wilków | 16 | 15 | 8 | 14.000 | 0.992 | 3.950 |
| Krzywa Góra | 39 | 30 | 13 | 13.078 | 0.981 | 3.620 |
| Nart | 35 | 29 | 16 | 13.790 | 0.990 | 3.369 |
| Mean | 34.429 | 26.429 | 12.571 | 13.170 | 0.982 | 3.097 |

AMOVA analysis of genetic variation demonstrated the presence of significant genetic differentiation between the studied sites for populations in both F1 ($p_{value}$ = 0.000) and F2 ($p_{value}$ = 0.006). The intra-site variation coefficient was significant for F1 and F2 (Table 3, Figure 2) and the percentage of intra-site variation in F1 was more than 0.86, and for F2, it was almost additionally 0.11 higher (Table 3). The Mantel analysis [30] did not demonstrate any relationship between geographical and genetic distances for both F1 ($p_{value}$ = 0.978) and F2 ($p_{value}$ = 0.699) (Table 3, Figures S1 and S2, Supplementary Materials).

**Table 3.** AMOVA and IBD Mantel [30] results for the two analysed generations of stands.

| Source of Variation | d.f. | Sum of Squares | Variance Components | Percentage of Variation |
|---|---|---|---|---|
| | | F1 generation | | |
| Among populations | 6 | 147.358 | 0.22803 | 13.17 *** |
| Among individuals within populations | 324 | 974.007 | 1.50310 | 86.83 *** |
| Within individuals | 0 | 0.000 | 0.00000 | 0.00 |
| Total | 661 | 1121.366 | 1.73112 | |
| IBD Mantel | | | R | *p*-value |
| | | | 0.0064 | 0.978 |
| | | F2 generation | | |
| Among populations | 6 | 34.026 | 0.04251 | 2.96 ** |
| Among individuals within populations | 234 | 651.633 | 1.39238 | 97.04 *** |
| Within individuals | 241 | 0.000 | 0.00000 | 0.00 |
| Total | 481 | 685.660 | 1.43489 | |
| IBD Mantel | | | R | *p*-value |
| | | | −0.0897 | 0.6991 |

Statistical significance of *p*-value ** < 0.01; *** < 0.001.

The value obtained for genetic distance ($F_{st}$) between F1 locations ranged from 0.00786 to 0.25852 and was considered statistically significant in most cases (Table 4). A lack of genetic distance was detected for two F1 pairs: Granica and Czerwińskie Góry ($p_{value}$ = 0.25586) and Wilków and Sieraków ($p_{value}$ = 0.55176) (Table 4). In F2, the diversity coefficients ranged from 0.0018 to 0.11074 (Table 4), and in 0.47 of cases were not considered genetically diverse in terms of statistical significance. The clustering of analysed locations for F1 and F2 generations allows the visualisation of F2 aggregation into a separate subgroup relative to all locations belonging to the maternal generation (Figure 3).

**Table 4.** The genetic differentiation (*F*st) between populations analysed (below diagonal) and their statistical significance (*p* values) (above diagonal) results in bold are not statistically significant.

| | SierakowF1 | SierakówF2 | Cz. GoryF1 | Cz. GóryF2 | GranicaF1 | GranicaF2 | WierszeF1 | WierszeF2 | WilkowF1 | WilkówF2 | Krzywa GóraF1 | Krzywa GóraF2 | NartF1 | NartF2 |
|---|---|---|---|---|---|---|---|---|---|---|---|---|---|---|
| SierakówF1 | | 0.00098 | 0.00000 | 0.01367 | 0.00000 | 0.04102 | 0.00000 | 0.07227 | **0.55176** | 0.01953 | 0.07910 | 0.00000 | 0.00098 | 0.71289 |
| SierakówF2 | 0.07608 | | 0.00000 | 0.00684 | 0.00000 | 0.00293 | 0.00000 | 0.00000 | 0.00000 | 0.00586 | 0.00000 | **0.14746** | 0.00000 | 0.00000 |
| Cz. GóryF1 | 0.23506 | 0.32715 | | 0.00000 | **0.25586** | 0.00000 | 0.00000 | 0.00000 | 0.00000 | 0.00000 | 0.00000 | 0.00000 | 0.00000 | 0.00000 |
| Cz. GóryF2 | 0.03988 | 0.06253 | 0.33028 | | 0.00000 | **0.85645** | 0.00000 | **0.22168** | 0.00879 | **0.69531** | 0.00000 | **0.65137** | 0.00000 | 0.00098 |
| GranicaF1 | 0.22296 | 0.32722 | 0.01388 | 0.33303 | | 0.00000 | 0.00000 | 0.00000 | 0.00000 | 0.00000 | 0.00000 | 0.00000 | 0.00000 | 0.00000 |
| GranicaF2 | 0.02662 | 0.06792 | 0.32038 | 0.00180 | 0.31591 | | 0.00000 | **0.29883** | 0.00781 | **0.41797** | 0.00000 | **0.18848** | 0.00000 | 0.00879 |
| WierszeF1 | 0.10276 | 0.23585 | 0.14030 | 0.22757 | 0.12369 | 0.21324 | | 0.00000 | 0.00000 | 0.00000 | 0.00195 | 0.00000 | 0.00000 | 0.00000 |
| WierszeF2 | 0.02614 | 0.09994 | 0.28641 | 0.02056 | 0.28288 | 0.01585 | 0.19672 | | 0.00781 | **0.19238** | 0.00000 | 0.00195 | 0.00000 | 0.00195 |
| WilkówF1 | 0.00786 | 0.07290 | 0.25287 | 0.04376 | 0.24788 | 0.04082 | 0.11451 | 0.03907 | | 0.01562 | 0.03125 | 0.00000 | 0.00488 | **0.25977** |
| WilkówF2 | 0.06101 | 0.09049 | 0.35240 | 0.01116 | 0.35023 | 0.02159 | 0.25149 | 0.03414 | 0.06114 | | 0.00000 | **0.28613** | 0.00000 | 0.00488 |
| Krzywa GóraF1 | 0.02506 | 0.13552 | 0.22874 | 0.10931 | 0.21859 | 0.09545 | 0.06186 | 0.09813 | 0.03079 | 0.11282 | | 0.00000 | 0.00488 | **0.09473** |
| Krzywa GóraF2 | 0.05783 | 0.03013 | 0.34615 | 0.00709 | 0.35239 | 0.01795 | 0.25523 | 0.04897 | 0.05840 | 0.03121 | 0.13168 | | 0.00000 | 0.00000 |
| NartF1 | 0.05378 | 0.17311 | 0.25852 | 0.15318 | 0.23628 | 0.13939 | 0.09328 | 0.14344 | 0.04390 | 0.16928 | 0.04168 | 0.16777 | | **0.07812** |
| NartF2 | 0.00611 | 0.11074 | 0.26168 | 0.06494 | 0.24697 | 0.04748 | 0.11945 | 0.05946 | 0.01751 | 0.08519 | 0.02894 | 0.08120 | 0.02831 | |

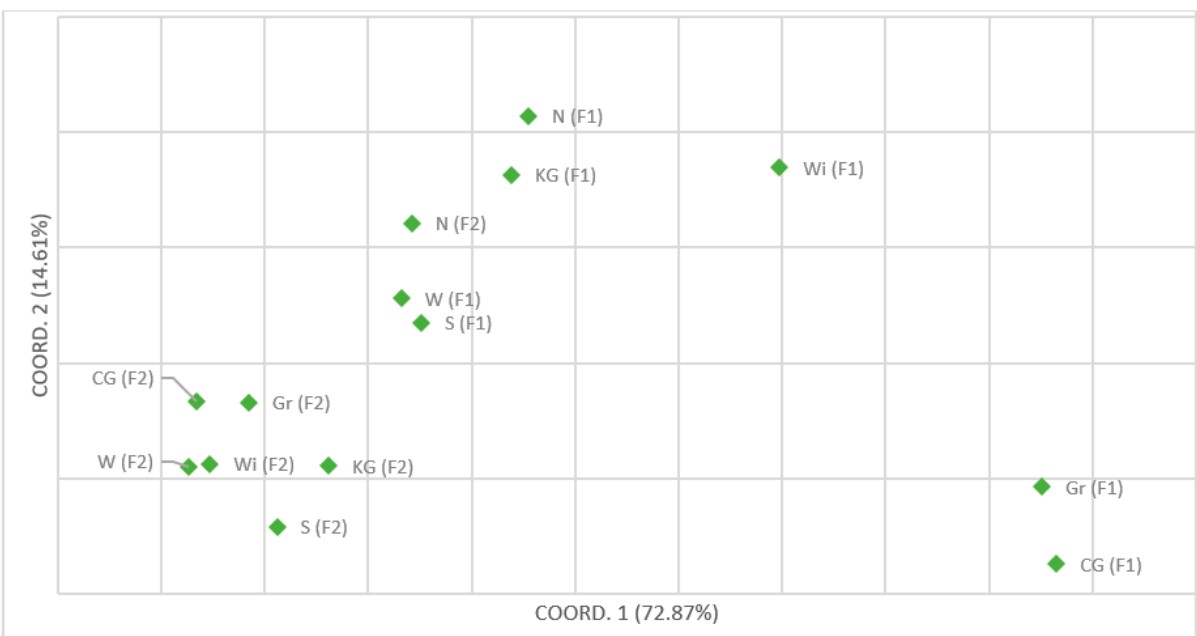

**Figure 3.** Grouping of studied Scot pine locations using principal coordinates analysis (PCoA) based on Nei genetic distance [32].

## 4. Discussion

The results of the presented study show that the maternal generation (F1) has a higher number of alleles than the progeny generation (F2). This result is further confirmed by an analysis of the effective number of alleles, the value of which is higher in the F1 generation, and the phenomenon is particularly evident for the S population. Detailed visualisation of the haplotype frequencies for F1 and F2 locations confirmed not only the depletion of the gene pool in F2 but also the presence of haplotypes foreign to F1. This tendency cannot be explained by the presence of private alleles only in F1, which could be due to individual selection during the growth phase of the stand [33], as the mean value of private alleles is identical between generations. The observed differences in the numbers of private alleles are only observed at the level of individual sites, which is evident in the example of stand S, where the value of private alleles in F2 was much lower compared to F1. The observed phenomenon of reductions in the allele numbers for other locations in the F2 generation can be explained by the preference for certain alleles during the reproductive process, which may facilitate population adjustments [34]. In the presented analyses, a more viable cause of changes in the frequency and occurrence of haplotypes may be due to the old forests being contaminated with pollen from younger, surrounding stands. KNP stands younger than those examined in the present study were planted for regeneration after war damage. Their genetic pool is therefore probably poorer in terms of diversity, and intensive pollen production is important for the creation of new generations of KNP. The effect of pollen flow between stands has been described by Lindgren et al. [35], Harju and Nikkanen [36], and Adams and Burczyk [37], among others, who confirm its role in forming the allele pools of the next generation of stands. Detailed analyses of haplotypes confirmed significantly higher values of haplotype richness ($R_h$) in the F1 generation of the studied sites. The values obtained for both the $R_h$ and $D^2sh$ parameters were almost twice as high in the F1 generation as in F2 generation. On the other hand, it should be noted that there are no clear differences in the values of genetic variability ($H_e$) between the studied generations. The obtained $H_e$ results were high for both F1 and F2, which is a typical phenomenon in conifers [38]. The results demonstrate significant differentiation at the studied sites, both between and within populations. Typically for conifers, the studied stands show the largest amount of within-population genetic variation, 0.86 for F1 and 0.97 for F2. Notably, the observed variation between populations showed significance of the difference at the level

of $p = 0.001$ for F1 and $p = 0.01$ for F2. The IBD–Mantel analysis [30] did not indicate any significant relationship between the values of genetic and geographical distances for the studied sites. Effective pollen transfer between the study sites should therefore be ruled out. The obtained results confirm those discussed above, and we can assume that the emergence of genetic richness was dominated by foreign pollen, probably originating from younger stands in the vicinity. However, this hypothesis requires further investigation, especially in the context of the function of national parks for biodiversity conservation at the genetic level. On 21 January 2000, KNP was recognised by the International Coordinating Council of the UNESCO MaB (Man and Biosphere) Programme as a "Puszcza Kampinoska" Biosphere Reserve with an area of 76 232.57 ha. Since 2004, a fragment of KNP with an area of 37,469.70 ha, which includes the central and buffer zones, has been part of the European Natura 2000 network as a Natura 2000 site with the number PLC 140,001 (https://www.kampinoski-pn.gov.pl/ochrona-przyrody/natura-2000, accessed on 10 April 2021) due to its high bird species richness and unique diversity of plant communities. Ensuring the stability of biodiversity, including at the genetic level, is an important aim for KNP. Considering the results obtained, it is likely that the implementation of total protection for the areas analysed in the project will result in the loss of private alleles. According to some predictions, private alleles may be carriers of genetic variation that allow populations to adapt to a changing environment in the future, even though their current impact on populations may be marginal [39,40]. For the above reasons, we recommend rare genotypes at selected naturally valuable sites be conserved through an ex situ conservation method. Under legally protected natural conditions, it may not be possible to maintain a unique set of alleles for ecologically valuable stands. The genetic distances determined for the studied sites were significant for F1, and the evaluation for F2 indicated that 47% of sites have lost their individual character. The grouping of the studied sites, as presented, illustrates the unification of the genetic variation of F2 in relation to the studied F1. This result confirms the loss, through natural processes, of rare genetic information that determines the individual character of the studied ecosystems and their high ecological value.

## 5. Conclusions

Significant differences in revealed haplotypes between the analysed generations were shown at the studied sites in the present analyses. Significant genetic differences between sites were also demonstrated, but it should be noted that the genetic distance decreased significantly in the next stand generation. In addition, the studied sites showed higher allelic diversity in F1, but this was not maintained in the subsequent stand generation. It is suggested that this is due to the high influence of site contamination by foreign pollen. However, determining whether this is indeed the cause requires further research. The obtained results indicate the need to maintain, ex situ, the genetic variability of naturally valuable old forest stands.

**Supplementary Materials:** The following are available online at https://www.mdpi.com/article/10.3390/d14020093/s1, Tables S1–S8: Geographical coordinates of F1 and F2 plant material from the analysed locations; Figures S1 and S2: Correlation of geographical and genetic distance in IBD Mantel [30] for F1 and F2.

**Author Contributions:** Conceptualization, P.P.; methodology, P.P., A.T. and J.M.; software, P.P.; validation, P.P. and A.T.; formal analysis, P.P., A.T. and J.M.; investigation, P.P.; resources, P.P.; data curation, P.P.; writing—original draft preparation, P.P.; writing—review and editing, P.P., A.T., J.M. and Ł.T.; visualization, P.P. and J.M.; supervision, P.P., I.S.-B; project administration, P.P.; funding acquisition, Ł.T. All authors have read and agreed to the published version of the manuscript.

**Funding:** This research was funded by Polish State Forest in 2020–2021.

**Institutional Review Board Statement:** Not applicable.

**Informed Consent Statement:** Not applicable.

**Data Availability Statement:** Project data publicly available in the Forest Research Institute library.

**Conflicts of Interest:** The authors declare no conflict of interest.

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
