# Peer review of "Conservation of Genetic Diversity of Scots Pine (Pinus sylvestris L.) in a Central European National Park Based on cpDNA Studies"

_diversity, doi:10.3390/d14020093_

Round 1

Reviewer 1 Report

This small study nicely compares the genetic diversity in two generations of a scots pine stand. However, several sections of the manuscripts can be improved. For example, firstly, the introduction section is relatively poor and does not identify knowledge gaps or justify the study- it should be improved. Second, it would be nice to map the sampling locations (currently presented as coordinates in the table). This is because understanding spatial separation of sampling location may help evaluate some of the authors' hypotheses to describe their results (e.g., foreign pollen arriving from the vicinity of the stands).  

Author Response

Thank you for the very information concerning the improvement of my proposed article.
I accept without question the need for a deeper analysis of the literature. I have added to the introductory information of the chapter on the research problem to be analysed. I have added precise information about variation among populations and within populations of the species in lines 64-74. The added articles discuss the raised issue of the influence of external pollen transmission, which is also relevant in this publication. A manuscript on the presentation of a complete chloroplast genome for Pinus uliginosa (Neumann) was also indicated. As suggested, a map showing the exact locations of the study plots was included with the manuscript.

Reviewer 2 Report

The manuscript “Conservation of genetic diversity of Scots pine (Pinus sylvestris L) in a central European national park based on cpDNA studies” by Przybylski et al. aims to address the study of levels of genetic diversity in populations of two generations of P. sylvestris growing in natural conditions and excluded from forest management. The populations of long-lived forest tree species have a structure with overlapping generations and the studies of successive generations allow for the inference of proceeding demographic processes. Knowledge of the genetic structure dynamics of the forest trees is of importance for management practices and conserving species adaptive potential in a changing world.

The work has clear aims, the research was conducted competently, the results are quite interesting but there are a series of significant changes that should be made. The presentation should be improved considerably before the paper can be published. 

General comments

The authors studied populations of F1 and F2 generations in seven locations. It seems that it is better to change the word  ‘location’ to ‘population’ throughout the text.

The Introduction section of the Article is rather limited and does not cover all published studies on the subject under study (e.g., Wojnicka-Półtorak et al., 2017, 2018). 

In the Materials and Methods section, DNA extraction method, and a scroll of chloroplast microsatellite markers used are omitted.

The Data analysis section does not give the method used for grouping of populations studied.

Results

It will be helpful to provide the parameters of genetic diversity obtained with chloroplast microsatellite loci for populations studied in the form of a table in the supplementary materials.

The authors should indicate how many haplotypes were determined in each generation and in both generations. Did you find any common haplotypes in two different generations or in different individuals of each generation?

Figure 1 is poorly perceived, the distribution of haplotypes in the studied populations is hard to follow, and even more so the frequency of their occurrence. It is not clear what the percentages on the Y-axis indicate (000; 000; 001 ...). There are too many colors - colors should aid the visual distinction of different haplotypes, but with such large quantities of haplotypes being similar it is hard.

I hope the authors will find the following specific comments helpful in the revision:

  1. Line 30, please add the reference (1) in the sentence with percentages of the area covered.
  2. Lines 44-45, please rephrase the sentence ‘The analyses presented in this paper in relation to previous research [6] could be adaptive in nature.‘, this wording is awkward and the sense of the sentence is unclear.
  3. Line 56–58, please rephrase ‘with little work in populations that are currently in dynamic equilibrium.‘, the wording is awkward 
  4. Line 83–85, please rephrase the sentence ‘The study was conducted in Kampinos National Park, a location dominated by Scots stands in the upper layer, which is located in strictly protected areas and excluded from forest management’, sense of the sentence is unclear
  5. Line 87, table 1 is poorly formatted. Please, reformat the Table and explain what the asterisk means. Consider please, maybe swap columns and rows.
  6. Line 90, what is the meaning of the ‘Plant material ... in the form of needles’, rephrase, please. It seems that this wording is awkward, which should need to revise.
  7. Lines 98–99, what is the meaning of ‘Properly prepared, protected, and described material...’ This should be described properly.
  8. Line 155, Legend of Figure 1. The mean of this Legend is confusing. Please revise it carefully.
  9. Line 162, please change ‘plot location’ to ‘geographical’ and ‘for both F1 (P value = 0.978) and F2 (P value = 0.699)’ to ‘for populations in both F1 (P value = 0.978) and F2 (P value = 0.699)’.
  10. Lines 164 and 166, please change ‘genetic diversity’ to ‘genetic distance’.
  11. Line 168. How many genetic distances between populations of F2 generation are insignificant? You write ‘0.47 cases’ on Line 168 and ‘45%’ on Line 234.
  12. Line 169–171, please, change ‘analysed locations for F1 and F2 generations’ to ‘analyzed populations of F1 and F2 generations’ and ’all locations’ to ‘all populations’    
  13. Line 172 and throughout the text, Change ‘Mentel’ to ‘Mantel’
  14. Line 176, Legend of Figure 2. Please denote, the results of what analysis are shown in this figure, and remove the reference to the software package for multiple analyses. The references to the software packages and analyses should be described in detail in Data analyses section.
  15. Line 177, please correct the title of table 4. For example, ‘The genetic differentiation (Fst) between populations analysed (below diagonal) and their statistical significance (P values) (above diagonal); P values < 0.05 are shown in bold.’ Please consider if it is better to highlight in bold the insignificant values of genetic distances, rather than P > 0.5 for some genetic distance values. 
  16. Line 182–185, please rephrase the sentence ‘Detailed visualization of the haplotype frequencies for F1 and F2 localisations confirmed not only the depletion of the gene pool in F2 but also its qualitative change compared to the haplotypes observed for the localisations’. The mean of this sentence is confusing. Please revise it.
  17. Line 240, please change ‘in  generated haplotypes’ to ‘in revealed haplotypes’
  18. References: please check reference numbering;

reference #11 please change authors names to ‘O'Grady J.,  Brook B.,  Reed D.,  Ballou J.,  Tonkynv D.,  Frankham R.’ and add a year to this referance;

reference Razor et al. 2019,   please add an article title to this referance

reference #20, please correct the article title

  1. Figures 1S and 2S, please remove lettering in Russian and change it to English ones

Author Response

The manuscript “Conservation of genetic diversity of Scots pine (Pinus sylvestris L) in a central European national park based on cpDNA studies” by Przybylski et al. aims to address the study of levels of genetic diversity in populations of two generations of P. sylvestris growing in natural conditions and excluded from forest management. The populations of long-lived forest tree species have a structure with overlapping generations and the studies of successive generations allow for the inference of proceeding demographic processes. Knowledge of the genetic structure dynamics of the forest trees is of importance for management practices and conserving species adaptive potential in a changing world.

The work has clear aims, the research was conducted competently, the results are quite interesting but there are a series of significant changes that should be made. The presentation should be improved considerably before the paper can be published. 

I would like to thank the reviewer for a very thorough and careful proofreading of my work. The review provided has clearly improved its content. I have implemented all the reviewer's comments as stated below. 

General comments

The authors studied populations of F1 and F2 generations in seven locations. It seems that it is better to change the word  ‘location’ to ‘population’ throughout the text.

I do not agree with this proposition, which is due to the ambiguity of the word population. Population may be the Kampinos forest as a whole. Place specifies the location of the investigation. Reformatting the word place into population may lead to a misperception of the text.

The Introduction section of the Article is rather limited and does not cover all published studies on the subject under study (e.g., Wojnicka-Półtorak et al., 2017, 2018). 

I accept without question the need for a deeper analysis of the literature. I have added to the introductory information of the chapter on the research problem to be analysed. I have added precise information about variation among populations and within populations of the species in lines 64-74. The added articles discuss the raised issue of the influence of external pollen transmission, which is also relevant in this publication. A manuscript on the presentation of a complete chloroplast genome for Pinus uliginosa (Neumann) was also indicated.

In the Materials and Methods section, DNA extraction method, and a scroll of chloroplast microsatellite markers used are omitted.

Missing DNA extraction data were completed in lines 128-132. The lack of description of the markers used is due to reference to the literature in which the same set was used [23-25].

The Data analysis section does not give the method used for grouping of populations studied.

The grouping of populations for analysis was based on geographical location. This fact is described in chapter 2.2. In the context of genetic analyses, gruping is described by lines 153-155.

Results

It will be helpful to provide the parameters of genetic diversity obtained with chloroplast microsatellite loci for populations studied in the form of a table in the supplementary materials.

The results were added to the supplementary material, the information was annotated in manuscript line 168.

The authors should indicate how many haplotypes were determined in each generation and in both generations. Did you find any common haplotypes in two different generations or in different individuals of each generation?

The information mentioned by the reviewer is included in the manuscript in Table 2 and described in lines 170-182. If additional information is needed, please provide details.

Figure 1 is poorly perceived, the distribution of haplotypes in the studied populations is hard to follow, and even more so the frequency of their occurrence. It is not clear what the percentages on the Y-axis indicate (000; 000; 001 ...). There are too many colors - colors should aid the visual distinction of different haplotypes, but with such large quantities of haplotypes being similar it is hard.

Thank you for calling attention to Figure 1, now Figure 2. Its main purpose is to illustrate to the reader the variation in haplotypes between generations. It was not the authors' intention to force the reader to do a thorough colorimetric analysis. The reviewer is correct that the legend could have been controversial, so we have omitted it and added additional notes in the description. The description of the OY axis has been corrected.

I hope the authors will find the following specific comments helpful in the revision:

  1. Line 30, please add the reference (1) in the sentence with percentages of the area covered.

It has been added in accordance with the reviewer's recommendations

  1. Lines 44-45, please rephrase the sentence ‘The analyses presented in this paper in relation to previous research [6] could be adaptive in nature.‘, this wording is awkward and the sense of the sentence is unclear.

The sentence was rephrased

  1. Line 56–58, please rephrase ‘with little work in populations that are currently in dynamic equilibrium.‘, the wording is awkward 

The sentence was rephrased

  1. Line 83–85, please rephrase the sentence ‘The study was conducted in Kampinos National Park, a location dominated by Scots stands in the upper layer, which is located in strictly protected areas and excluded from forest management’, sense of the sentence is unclear

The sentence was rephrased

  1. Line 87, table 1 is poorly formatted. Please, reformat the Table and explain what the asterisk means. Consider please, maybe swap columns and rows.

The missing information has been filled in, the table format is homogeneous with previous publications as "Przybylski, P.; Mohytych, V.; Rutkowski, P.; Tereba, A.; Tyburski, Ł.; Fyalkowska, K. Relationships between Some Bio-diversity Indicators and Crown Damage of Pinus sylvestris L. in Natural Old Growth Pine Forests. 2021 Sustainability 13,1239, https://doi.org/10.3390/su13031239."

  1. Line 90, what is the meaning of the ‘Plant material ... in the form of needles’, rephrase, please. It seems that this wording is awkward, which should need to revise.

The sentence was rephrased

  1. Lines 98–99, what is the meaning of ‘Properly prepared, protected, and described material...’ This should be described properly.

The sentence was rephrased

  1. Line 155, Legend of Figure 1. The mean of this Legend is confusing. Please revise it carefully.

Thank you for calling attention to Figure 1, now Figure 2. Its main purpose is to illustrate to the reader the variation in haplotypes between generations. It was not the authors' intention to force the reader to do a thorough colorimetric analysis. The reviewer is correct that the legend could have been controversial, so we have omitted it and added additional notes in the description. The description of the OY axis has been corrected.

  1. Line 162, please change ‘plot location’ to ‘geographical’ and ‘for both F1 (P value = 0.978) and F2 (P value = 0.699)’ to ‘for populations in both F1 (P value = 0.978) and F2 (P value = 0.699)’.

Revised in accordance with the reviewer's recommendations

  1. Lines 164 and 166, please change ‘genetic diversity’ to ‘genetic distance’.

Revised in accordance with the reviewer's recommendations

  1. Line 168. How many genetic distances between populations of F2 generation are insignificant? You write ‘0.47 cases’ on Line 168 and ‘45%’ on Line 234.

Thank you for drawing attention to this inconsistency. Such an error should not have occurred to me, for which I apologise. I have homogenised the result.

  1. Line 169–171, please, change ‘analysed locations for F1 and F2 generations’ to ‘analyzed populations of F1 and F2 generations’ and ’all locations’ to ‘all populations’

As I explained earlier, if it is possible I would ask that the word location be left out. The word has also been used in previous publications without causing controversy.

  1. Line 172 and throughout the text, Change ‘Mentel’ to ‘Mantel’

The error has been corrected as suggested by the reviewer

  1. Line 176, Legend of Figure 2. Please denote, the results of what analysis are shown in this figure, and remove the reference to the software package for multiple analyses. The references to the software packages and analyses should be described in detail in Data analyses section.

The error has been corrected as suggested by the reviewer

  1. Line 177, please correct the title of table 4. For example, ‘The genetic differentiation (Fst) between populations analysed (below diagonal) and their statistical significance (P values) (above diagonal); P values < 0.05 are shown in bold.’ Please consider if it is better to highlight in bold the insignificant values of genetic distances, rather than P > 0.5 for some genetic distance values

The title has been corrected on the basis of a reviewer's suggestion

  1. Line 182–185, please rephrase the sentence ‘Detailed visualization of the haplotype frequencies for F1 and F2 localisations confirmed not only the depletion of the gene pool in F2 but also its qualitative change compared to the haplotypes observed for the localisations’. The mean of this sentence is confusing. Please revise it.

A sentence has been corrected

  1. Line 240, please change ‘in  generated haplotypes’ to ‘in revealed haplotypes

A sentence has been corrected

  1. References: please check reference numbering;

reference #11 please change authors names to ‘O'Grady J.,  Brook B.,  Reed D.,  Ballou J.,  Tonkynv D.,  Frankham R.’ and add a year to this referance

A reference has been corrected

reference Razor et al. 2019,   please add an article title to this referance

A reference has been corrected

reference #20, please correct the article title

A reference has been corrected

  1. Figures 1S and 2S, please remove lettering in Russian and change it to English ones

Corrected

Reviewer 3 Report

Manuscript Number:  diversity-1506030

Title:   Conservation of genetic diversity of Scots pine (Pinus sylvestris L.) in a central European national park based on cpDNA studies

Authors:  Paweł Przybylski, et al.

Remarks:

This submitted paper compared genetic diversity of mother and progeny populations of Scots pine and tried to consider the conservation of genetic variation over generations. The objective itself is important, however, I consider that the present paper do not reach the level of the publication as a research paper, because of several problems, such as verbose writing style such as unnecessary sentences and statements, unclear and irrelevant presentations particularly in Introduction and Discussion, and incorrect word uses and English writing. The present version is far from the understandings by the readers on what, how and why the authors have performed the study with a relevant logic, procedure and indication.

I recommend the authors to reconsider the entire manuscript with a sufficient effort and resubmit, based on the major comments as below.

Major Comments:

  1. The main problem of the present version is that most of the sentences and presentations in the paper is unclear and insufficient, and I cannot understand what they mean by. It is partly because of the lack of backgrounds and reasons why they state so, particularly in Introduction. For example, there is no clear explanation of “natural processes affecting the genetic diversity of pine populations” (Lines 34-), “with little work in populations that are currently in dynamic equilibrium” (Lines 58-), and “evolution did not occur by natural regeneration” (Lines 70). I think that the reason of “the increasing importance of cpSSR analyses” (Lines 62-) is necessary. Statements around these unclear and insufficient words relate strongly with the main parts of the Introduction, therefore, unless these parts cannot be understood by the readers, no one cannot appreciate the relevance and the value of the study. The Discussion section is also rather unclear throughout; I cannot understand what the authors want to say based on the obtained results. So, the present paper needs the entire revisions by the authors, also including Methods, Results, to make clear the backgrounds of how and why the authors targeted the study point, and to show the significance of the study. Please reconsider.

  1. Explanation of the materials is unclear; what the maternal generation (F1-population) was derived from? Does “F1” mean the first-generation breeding population; an assemblage of the selection of trees from the outside forests (original populations)? These trees are the main material of the present study; therefore, unless the explanation is unclear, the readers cannot appreciate the relevance and significance of the whole study. I also recommend the authors to provide a figure to visualize the distributions of the study materials together with Table 1. Please reconsider.

  1. Some of results may be problematic; I don’t imagine why the F2 populations, which are expected to have lower and skewed genetic variations compared with each of their F1 populations, has low genetic difference (similar genetic compositions) among populations and aggregated to the nearer positions in PCoA. In fact, I suspect that the F2-populations are derived from the common (or some of the genetically related) seed pools, not derived from each corresponding F1-populations. How the authors interpret the results? The authors should discuss about the results with more relevant reasons and backgrounds so that the readers can appreciate the value of the study.

  1. In Discussion section also, there are some incorrect interpretations of the results. For example, in Lines 195-, “KNP stands younger than those … after war damage.”, it is irrelevant unless the effect of “war damage” on the genetic variation of the species has been recognized, suggested or discussed in advance. Also, the interpretation of “It is suggested that … contamination by foreign pollen.” is not based on the previous findings at all and therefore seems irrelevant. The authors should provide the discussions about the background findings or suggestions in order to support the authors’ statements.

  1. There are a lot of unnecessary and redundant sentences throughout the manuscript, which leads to verboseness of the paper. For example, in Introduction, the sentence “The present study was … (KNP).” (Lines 37-) is almost the same as the sentences of Lines 67- “The research presented … in Poland.” and also of Lines 83- (the first sentence in the M&M section). In Table 3, the column of “Within individuals” is unnecessary. Merely an array of the pairwise FST values in Table 4 seems rather unclear; Fig. 2 would be sufficient to show the genetic differences clearly for the readers. The authors should reconsider the entire manuscript more carefully to represent the study more effectively and catchy by the readers.

  1. There are a lot of incorrect and unclear word uses which should be refined, and also use of the words should be uniform. For example, I do not understand what the authors mean by the words “the transmission of pine genetic information by seeds and pollen” (Lines 33-). The words “plot location” (Lines 162-) should be “geographical distance”. Saying the individual trees (samples), there are various words; “tree”, ”individual” and ”pine”; which should be uniformed. Regarding with this, the English is not refined entirely throughout the manuscript. Didn’t the authors process the English-editing before this time of the submission? If not, the authors should process after the entire revisions of the contents of the paper, before the next time of submission.

Author Response

Dear Reviewer

I would like to address the criticism of the manuscript in two dimensions: general and specific.
 In the general context, the reviewer pointed out the inaccuracy of the statement and the need for a thorough (major) revision of the manuscript. This conclusion is at odds with the reviews received of the current version of the paper. I respect the reviewer's opinion, but disagree with it. The text was proofread by MDPI (for which I hold a certificate) for language and accuracy of expression. The resubmission of the text for linguistic proofreading therefore puzzles me. 
In a detailed context, the reviewer asks about several main issues.
The F1 and F2 generations are described in section 2.2 lines 122-134. F1 trees are the maternal generation, F2 seedlings the offspring generation (useful symbols and explained in the text). The actual population distribution has been added to the manuscript. 
The reviewer's assumption of a common F2 origin is erroneous and incomprehensible. The collection of material is discussed in detail in section 2.2. The reduction in genetic distance between populations is interesting and probably due to homogenization of the gene pool by transfer of foreign pollen. This is also discussed in the Wojnicka-Półtorak 2017 study appended to the manuscript. This aspect is presented both in the discussion and in the conclusion of the submitted work.
The reviewer has doubts in connection with the discussion of pollen transfer from younger, surrounding analysed populations of stands. Their origin is not clear and pollen transfer is significant according to the authors. In the discussion chapter, I may present my opinion in the context of the results obtained. A similar tone of analysis is struck in the referenced studies. 
I do not understand why the reviewer has doubts about the transfer of genetic information by seeds or pollen, this is a dogma of biology! The locations of the plots have been included in the text and the exact geographical coordinates of each tree are listed in the supplementary material.
In conclusion, I am sorry to say that the proposed review, which requires a thorough revision of the text, is not possible. Based on the advice I received from the other reviewers, I have revised the text and hope that it will receive a favourable review. Thank you for reading the manuscript and please provide me with detailed comments to really improve the manuscript.
 Yours sincerely

Paweł Przybylski

Round 2

Reviewer 1 Report

Thanks for addressing my comments and careful revision. 

Author Response

Dear Reviewer

I am very grateful for the work done to improve the manuscript. All the comments made during the review helped to improve the quality of the manuscript. I am glad that the corrections made after your review are accepted and sufficient.

Best regards

Paweł Przybylski

Reviewer 2 Report

The Authors have put considerable effort into improving the presented manuscript, however, I do have some more corrections and editorial suggestions that may improve the manuscript. Below are a few items that should be addressed: 

  1. Line 60. Please change ‘populations of a natural character’ to ‘natural populations’.
  2. Line 96. Please change ", which are located in strictly protected areas" to "and located in strictly protected areas"
  3. Line 98. Please correct typing error ‘nad’ to ‘and’.
  4. Line 104. This figure may well illustrate the locations of the study areas in the Park but some questions are raised: What are the circles of different colors and sizes designated? Do they indicate protected areas or locations (populations) studied? Then where are located the Granica (Gr), Nart (N), Czerwińskie Góry (CG) pine stands in the Figure, why is no circle in the Sieraków location, and are there two circles in Krzywa Góra location? It may be better to indicate the names of locations of the Park and mark each location (population) studied with point and Abbreviation.
  5. Line 126. You write ‘Molecular analyses were performed using 6 chloroplast microsatellite markers [24-26].’  However in these articles, 20 [24], 17 [25], and 8 microsatellite loci [26] were described, and you have used only six microsatellite loci. Please indicate (by listing them in the Materials and Methods), which namely six chloroplast microsatellite loci from these articles did you use in this work?
  6. Lines 148-149. You write “The grouping of analyzed stands on the basis of the genetic distance of Nei was performed in the GenALEx 6.5 [296] program”, and represent the result in Figure 3. However, you should report what namely analysis from the GenALEx 6.5 program you have used. For example: 'was performed using Principal Coordinates Analysis (PCoA) as implemented in the GenALEx 6.5 [296] program.'
  7. Line 156–162. Please move the reference to Table 8S, for example, to line 156: ‘An average of 4.07 alleles were found in F1 generation populations (see Supplementary Materials, Table 8), with most in the S location (4.5) and the least in CG (3.7).’ The present position of the reference to Table S8 in the sentence on haplotype diversity does not correspond to the results presented therein.
  8. Are there common haplotypes in populations of the different generations from the same location or in populations of the same generation from different locations? or there are no common haplotypes at all? Could you indicate this in the text (for example in paragraph at 164-173 Lines)?
  9. Line 172. Please correct typing error ‘(D2sh)’ to (D2sh). 
  10. Line 181. Please change  the title of Figure 2: ‘between maternal (F1) and progeny (F2) generations‘ to ‘of maternal (F1) and progeny (F2) generations’ 
  11. Line 188. Please change ‘plot location and genetic distance’ to ‘geogra[hical and genetic distances’
  12. Line 203. Please change the title of Figure 3 to ‘Grouping of studied Scot pine locations using principal coordinates analysis (PCoA) based on Nei genetic distance [32] '
  13. Line 211. Please change ‘localisations’ to ‘locations’
  14. Please clarify the following references: #4- add vollume and pages: 9: 801−812.  doi: 10.3832/ifor1577-009; #11- change the order of words as in other references: 2006, Biological Conservation. 133: 42 – 51. doi.org/10.1016/j.biocon.2006.05.016; #24- qualify the title of this reference (see DOI: 10.1111/j.1365-294x.1996.tb00353.x); #38 - correct a typing error in article title: ‘gen’ to ‘gene’.
  15. Supplementary Materials. Figure 1S and 2S. Please remove or change the word 'Логарифмическая' in the figure

Author Response

Dear Reviewer

I am very grateful for the work you have done in reviewing the article. I have made all the suggested corrections to the text. Figure 1 has been changed according to the detailed guidelines in the review. The names of the markers in the materials and methods have been introduced and the text has been corrected as suggested. I believe that all the reviewers' comments have significantly improved the quality of the submitted scientific paper, for which I would like to express my sincere gratitude.

Best regards

Paweł Przybylski